# State of the Art in Constructing Gas-Propelled Dissolving Microneedles for Significantly Enhanced Drug-Loading and Delivery Efficiency

**DOI:** 10.3390/pharmaceutics15041059

**Published:** 2023-03-24

**Authors:** Minmin Zhang, Beibei Yang, Xuanyu Luan, Ling Jiang, Chao Lu, Chuanbin Wu, Xin Pan, Tingting Peng

**Affiliations:** 1School of Pharmaceutical Sciences, Sun Yat-Sen University, Guangzhou 510006, China; zhangminmin133@163.com (M.Z.); yangbb5@mail2.sysu.edu.cn (B.Y.); 2School of Chinese Materia Medica, Tianjin University of Traditional Chinese Medicine, Tianjin 301617, China; xuanyu.luan@outlook.com; 3Medical College, Shantou University, Shantou 515041, China; ljiang@stu.edu.cn; 4International Cooperative Laboratory of Traditional Chinese Medicine Modernization and Innovative Drug Development of Ministry of Education (MOE) of China, Jinan University, Guangzhou 511436, China; chaolu@jnu.edu.cn (C.L.); wuchuanb@mail.sysu.edu.cn (C.W.); 5College of Pharmacy, Jinan University, Guangzhou 511436, China

**Keywords:** gas-propelled microneedles, preparation technology, formulation optimization, drug loading, transdermal delivery efficiency

## Abstract

Dissolving microneedles (MNs) have emerged as a promising transdermal delivery system, as they integrate the advantages of both injection and transdermal preparations. However, the low drug-loading and limited transdermal delivery efficiency of MNs severely hinder their clinical applications. Microparticle-embedded gas-propelled MNs were developed to simultaneously improve drug-loading and transdermal delivery efficiency. The effects of mold production technologies, micromolding technologies, and formulation parameters on the quality of gas-propelled MNs were systematically studied. Three-dimensional printing technology was found to prepare male mold with the highest accuracy, while female mold made from the silica gel with smaller Shore hardness could obtain a higher demolding needle percentage (*DNP*). Vacuum micromolding with optimized pressure was superior to centrifugation micromolding in preparing gas-propelled MNs with significantly improved *DNP* and morphology. Moreover, the gas-propelled MNs could achieve the highest *DNP* and intact needles by selecting polyvinylpyrrolidone K30 (PVP K30), polyvinyl alcohol (PVA), and potassium carbonate (K_2_CO_3_): citric acid (CA) = 0.15:0.15 (*w*/*w*) as the needle skeleton material, drug particle carrier, and pneumatic initiators, respectively. Moreover, the gas-propelled MNs showed a 1.35-fold drug loading of the free drug-loaded MNs and 1.19-fold cumulative transdermal permeability of the passive MNs. Therefore, this study provides detailed guidance for preparing MNs with high productivity, drug loading, and delivery efficiency.

## 1. Introduction

In recent years, dissolving microneedles (MNs) have attracted increasing attention in the field of transdermal drug delivery, as they can painlessly penetrate the stratum corneum with higher delivery efficacy than conventional transdermal preparations [1,2,3,4]. However, the low drug-loading and limited delivery efficiency severely hinder the practical application of MNs. Particularly, it is difficult for high-dose drugs to attain effective drug loading, since the volume of needles is generally less than 10 µL, which is limited space for loading sufficient drugs [5]. To date, the drug delivery efficiency achieved from conventional MNs is still far from the effective therapeutic window concentration, as the drugs released from the MNs passively diffuse through the multilayer skin tissues ranging from the epidermis to the dermis, resulting in limited systemic absorption and low bioavailability [6,7].

Utilization of highly drug-concentrated particles to replace free drug solution during preparation has demonstrated an effective approach to improve the drug loading of MNs. This approach usually includes engineering of free drugs into highly drug-concentrated nanocrystals [8,9] or microparticles [10], and loading of the drug in the form of powder [11,12]. For drug-powder-carrying MNs, the drug powder was directly filled into the hollow cavity of the MNs and deposited at the tails of needles, which could decrease the drug delivery efficiency due to insufficient penetration into the skin by MNs [11,12]. In contrast, the drug microparticles eliminate the use of surfactants in large quantities associated with drug nanocrystals, and could be uniformly dispersed as a needle suspension to prepare MNs that do not affect the drug enrichment characteristics of needles.

A variety of physical and chemical permeation–enhancement techniques have been integrated with MNs to improve drug delivery efficiency. Common physical methods include infrared light, electric field, or ultrasonic irradiation. Pre-exposure of the skin to infrared light, electric fields, or ultrasound, followed by application of the MNs on the skin, or a synchronous treatment regimen, can effectively enhance the permeability of the drug [13,14,15,16,17,18]. However, infrared light and ultrasonic irradiation require external equipment and professionals to assist drug administration, which are complicated to operate, and there are a lack of quantitative studies between relevant parameters and penetration effects. Electric fields are only capable of delivering ionic compounds, and penetration of drug species and transdermal drug delivery remains limited. The combination of MNs and chemical permeation enhancers has also been reported to increase the permeability of drugs across skin. For example, amphoteric ionophores, degradative enzyme inhibitors, and vasodilators increase the amount of drug penetration by increasing the fluidity of cell membranes and the permeability of tissue interstitial spaces. However, a large number of chemical permeation enhancers are usually required to dramatically increase the transdermal delivery efficiency, which may lead to reduced MN mechanical strength, as well as vasodilators that can irritate blood vessels and increase the risk of skin irritation and safety issues.

At present, centrifugation micromolding and vacuum micromolding are two common technologies to pour the needle solution or suspension into the microholes of male mold for preparing needles [19,20,21]. However, centrifugation micromolding technology could cause the separation of polymer matrix and microparticles, as microparticles with higher density are prone to settle down into the microholes of mold during centrifugation. Asynchronous centrifugation would further cause the inhomogeneous distribution of needle compositions and the fracture of needles due to the poor mechanical nature of the microparticles. In contrast, vacuum micromolding technology could be a preferable alternative to centrifugation micromolding, as the former is driven by a pressure gradient that is mild for the synchronous settlement of needle compositions into the microholes of mold and hence could increase the formability of MNs [22]. Nevertheless, a systematic study on how the process and prescription factors affect the morphology and productivity of microparticle-embedded gas-propelled MNs is still lacking.

Based on above knowledge, the purpose of this study was to develop microparticle-embedded gas-propelled MNs to simultaneously enhance drug-loading and transdermal delivery efficiency (Figure 1). The optimization of preparation technologies and formulation compositions was conducted to obtain gas-propelled MNs with intact morphology and high productivity. Rivastigmine tartrate (RHT) was selected as the model drug to verify the feasibility of microparticle-embedded gas-propelled MNs for improved drug loading and delivery efficiency. Therefore, this study provides critical guidance for developing highly drug-loaded and permeated MNs that hold great promise for clinical applications.

## 2. Materials and Methods

### 2.1. Materials and Animals

Polyvinylpyrrolidone (PVP) K12, K30, K45, and K90 were obtained from Capricorn New Materials Technology Co., Ltd. (Shanghai, China). Citric acid (CA) was purchased from Sigma-Aldrich (Shanghai, China). PVA 1788 (MW 840000) was gained from McLean (Shanghai, China). Potassium carbonate and sodium carbonate were purchased from Tianjin Damao Chemical Reagent Factory (Tianjin, China). Hydroxypropyl cellulose (HPC-ELF) was purchased from (Ashland, Shanghai). PDMS was purchased from Dow Corning (Shanghai, China). Liquid silica gel was purchased from Dongguan Gangtian Polymer Materials Co., Ltd. (Dongguan, China). Dextran 70 (DEX) was purchased from McLean (Shanghai, China). Sodium hyaluronate (HA) was obtained from Focusfreda (China, Shandong). Rivastigmine hydrogen tartrate (RHT, purity > 99.8%) was obtained from Beijing Huawei Ruike Chemical Co., Ltd. (Beijing, China). The healthy male SD rats (220–250 g) and male C57BL/C female mice (16–20 g) were purchased from the Experimental Animal Center of Southern Medical University (License number: SCXK (Guangdong) 2021-0041). The animal experiment was reviewed by the Experimental Animal Ethics Committee of Sun Yat-sen University and complies with the relevant regulations of the National Experimental Animal Welfare Ethics. The approval number was SYSU-IACUC-2022-001871. The rats were raised in an SPF environment with a temperature of 25 ± 2 °C, a relative humidity of 40–70%, and an illumination of 15–20 lx.

### 2.2. Design and Preparation of Male Mold

Computer-aided design (AutoCAD, AutoDesk, San Rafael, CA, USA) software [23] was adopted to design the two-dimensional (2D) plane modeling of mold, and the SolidWorks mechanical design software (Dassault Systèmes, Paris, France) [24] was used to construct the three-dimensional (3D) modeling of mold that can ensure the accuracy and rationality of the geometric parameters during mold processing. The male mold was in conical shape with a needle tip length of 800 μm, the cone angle of needle set as 23°, and the diameter of the cone base set as 0.37 mm. The number of needles was set according to the actual male mold processing method.

The male mold of MNs was prepared by Micro Electro Mechanical Computer Numerical Control (MEMS-CNC) [19], 3D PuSL Projection Micro Stereolithography (3D PuSL) [23,25,26,27], UV laser drilling [5], and chemical etching [28,29], respectively, and compared in the morphology and geometric parameters of mold. The specific experimental parameters as well as the principles of the four preparation techniques are shown as follows.

(1)The MEMS-CNC method consists of fixing the brass material onto the high-precision machine tool (JDMR600, Beijing, China) and installing the milling cutter with a taper angle of 23° in the spindle tool slot. The program input is made according to the three-dimensional design drawing, and then the milling cutter is rotated at 15,000 rpm to mill the conical main mold.(2)The 3D-PuSL method consists of injecting the high-temperature-resistant photosensitive resin material into the resin tank through the pump liquid system of nanoArch^®^P140 3D printer (nanoArch ^®^ P140, Chongqing, China). The designed 3D model of MNs is imported into the high-precision ultraviolet lithography projection system, followed by instantaneous exposure to 405 nm UV light for curing. Then, the male mold of MNs is prepared by layer printing.(3)The ultraviolet laser drilling method uses a laser (FM-UVM5, Shanghai, China) with a wavelength of 355 nm, 65% output power (adjustable power range: 1–10 W), 150 mm/s laser speed (adjustable speed range: 100–200 mm/s), and a pulse frequency of 30 kHz to prepare MN mold on the surface of a 5 mm thick silicone plate by one-time burning.(4)In the etching method, the thick silicon nitride protective film is deposited on both sides of the silicon wafer by the low-pressure chemical vapor deposition (LPCVD) technology, and the photoresist is spun. The circular spot pattern of the mask is transferred to the photoresist to form the blocking adhesive film. Then, dry etching is carried out by inductively coupled plasma etching system. After cleaning up the treated silicon wafer with deionized water, isotropic wet etching is performed to obtain the MN mold.

### 2.3. Preparation of Female Mold and Optimization of Molding Materials

The female molds of MNs were prepared by inversely replicating the male mold. As displayed in Table 1, different types of molding materials containing silica gel and curing agent were slowly injected into the cavity of 3D-printed male molds. Then, the male molds were put into the vacuum-drying oven at −0.08 Mpa for 15 min to completely exclude the bubbles in the silica gel, followed by heating at 140 ± 5 °C for 25 min to cure the silica gel. Finally, the female molds of MNs were obtained by peeling out from the male mold.

To compare the influence of female molds prepared from different types of silica gel on the productivity of MNs, the dissolving MNs were prepared to calculate the demolding needle percentage (*DNP*) as an indicator of needle integrity. The *DNP* was calculated as the following Formula (1):(1)$$DNP\;\left(\%\right)=\frac{NT}{NS}\times 100\%$$
where *NT* represents the number of needles with intact morphology counted from the macroscopic images of MN arrays, and *NS* represents the number of needles based on the original design.

The dissolving MNs were prepared by a centrifugation casting method. Specifically, 150 μL of needle tip solution containing 0.08 g/mL PVP K30 and 0.25 g/mL HA was injected into the female mold, and centrifuged at 4 °C and 3500 rpm for 10 min. Then, the superfluous solution remaining on the surface of mold was removed. Furthermore, 0.5 mL of PVP K90 ethanol solution (0.26 g/mL) was injected into the female mold and centrifuged at 3500 for 5 min to prepare MN base. Finally, the MN mold was put in an electronic drying cabinet to dry for 24 h. The dissolving was then obtained to calculate the DNP.

To qualitatively correlate the relationship between female mold and DNP, the Shore hardness of silica gel was measured by Shore C hardness tester (LX-C, Beijing, China). Briefly, the silica gel mixture as displayed in Table 1 was further formulated into gel plates with a diameter of 3 cm and a thickness of 10 mm. Then, the Shore hardness of the gel plates was determined by the Shore C hardness tester.

### 2.4. Preparation of Gas-Propelled MNs and Passive MNs

#### 2.4.1. Preparation of RHT-Loaded Particles

First, 12 g of RHT was dissolved in 120 mL of deionized water, followed by sonication for 10 min to prepare RHT solution (10%, *w*/*v*). Then, 15 g of PVA was added into the RHT solution and swelled for 24 h to obtain the drug-loaded PVA solution. The drug-loaded PVA solution was further dried for 36 h at 55 °C and under a negative pressure of −0.08 Mpa. Finally, the dried mixtures were crushed and passed through 400-mesh sieve to obtain the RHT-loaded particles (PVA@RHT).

#### 2.4.2. Preparation of Needle Suspension and Base Solution

The RHT solution was prepared by dissolving 0.25 g RHT and 0.15 g citric acid (CA) in 3.0 g ethanol. Then, the polymers as described in Table 2 were added into the above ethanol solution for full swelling. Further, 0.15 g K_2_CO_3_ particles and 0.3 g PVA@RHT particles were dispersed in the polymer solution to form the needle suspension. The base solution of MNs was prepared by dissolving PVP K90 in the ethanol solution at a concentration of 33% (*w*/*v*).

#### 2.4.3. Preparation of Gas-Propelled and Passive MNs

To prepare the needles of MNs, 150 μL of the needle suspension was injected into the female mold of MNs, followed by vacuum micromolding or centrifugation micromolding. After microfilling, the excess needle suspension was scraped off from the surface of the female mold, which was further dried at 40 °C for 2 h. Then, 0.5 g of the base solution was added into the female mold and centrifuged at 4000 rpm for 10 min. Finally, the female mold was put in the glass dryer for drying. After drying for 12 h, the gas-propelled MNs were demolded from the female mold to determine the *DNP* and observe the morphology of MNs using optical microscopy (WIFI1000, Shanghai Renyue Electronic Technology Co., Ltd., Shanghai, China). The passive MNs were prepared by the same protocol of gas-propelled MNs, except for removal of the pneumatic initiators from the needle suspension.

### 2.5. Optimization of Gas-Propelled MNs

#### 2.5.1. Optimization of Needle Solution Microperfusion Method

For the vacuum micromolding, different vacuum pressures (−0.06 Mpa, −0.07 Mpa, −0.08 Mpa, −0.09 Mpa) were used to prepare the needles of gas-powered MNs. The female mold was placed under these vacuum-negative pressures for 2 min, and the cycle was repeated three times. For the centrifugation micromolding method, the needle suspension was poured into the micropores of female mold and centrifuged at 2500 rpm, 3000 rpm, 3500 rpm, or 4000 rpm for 5 min to prepare the needles of gas-propelled MNs. The other procedures were repeated as described above.

#### 2.5.2. Optimization of Polymer Materials as the Needle Skeleton

As described in Table 2, four kinds of polymers, including HPC, PVP K12, PVP K30, and PVP K45, were used as the needle skeleton materials to prepare gas-propelled MNs, and their effects on the productivity and morphology of MNs was also investigated.

#### 2.5.3. Optimization of Drug-Loaded Particles

The influence of drug-loaded particles, including drug carriers and the feeding concentration of drug particles, on the *DNP* and morphology of gas-powered MNs was also investigated. As displayed in Table 3, HA, DEX, and PVA were used as the carriers to prepare drug-loaded particles, and the feeding concentration of PVA@RHT particles was also optimized.

#### 2.5.4. Optimization of Pneumatic Initiators

The pneumatic initiators, including alkaline agents (K_2_CO_3_ or Na_2_CO_3_) and acid agents such as tartaric acid (TA) or citric acid (CA), were utilized to produce carbon dioxide (CO_2_) bubbles through acid–base reaction in water, thus accelerating the dissolution of MNs and enhancing local drug absorption. The detailed compositions of pneumatic initiators are shown in Table 4, and were investigated to prepare gas-powered MNs with good productivity.

### 2.6. Morphology of MNs Observed by Scanning Electron Microscope (SEM)

The morphology of MNs was observed by SEM (JSM-6330F, Jeol, Tokyo, Japan), which can provide detailed information on the surfaces, cracks, and deformations of the MNs.

### 2.7. Drug Loading and Distribution of MNs

The drug loading of MNs was determined by reverse-phase high-performance liquid chromatography (RP-HPLC, Waters 2695, Milford, MA, USA) at 214 nm. The mobile phase was composed of methanol and 0.05 M disodium hydrogen phosphate solution (58:42, *v*/*v*) and injected at the flow rate of 1 mL/min. The injection volume was 20 μL. Three kinds of microneedles, including free RHT-loaded passive MNs (F@RHT-MN), PVA@RHT loaded passive MNs (P@RHT-MN), and PVA@RHT coloaded gas-propelled MNs (P@RHT-GMN) were prepared. Moreover, rhodamine B (RhB) was used to replace RHT for preparing fluorescence-labeled MNs, which were further observed by laser confocal scanning electron microscopy (CLSM, Zessi, LSM 710, Jena, Germany) to visualize the drug distribution in the passive and gas-propelled MNs.

### 2.8. In Vitro and In Vivo Transdermal Permeability of MNs

The Franz diffusion instrument (Kaikai, TK-24BL, Shanghai, China) was used to conduct the in vitro transdermal permeability of MNs. Specifically, the passive and gas-propelled MNs were applied on freshly depilated skin that was further fixed to the Franz diffusion cell. Each receiving cell was filled with 7.5 mL of PBS buffer solution (pH = 5.8) as the dissolution medium. The transdermal permeation test was conducted at a stirring speed of 350 rpm and maintained at the temperature of 37 °C. At predetermined timepoint, 1 mL of receiving solution was taken from the diffusion cell and analyzed by HPLC to determine the content of RHT and calculate the cumulative percutaneous permeability.

To further study the passive and gas-propelled MNs, the RhB-labelled MNs were inserted into the excised mice skin, which was photographed by CLSM at 0.5 h and 1 h postadministration for observing the drug permeation in the skin tissues. Moreover, the RhB-labeled MNs were also applied to the living rat dorsal skin. At 4 h and 8 h postadministration, the mice were sacrificed to collect the skin tissue for tissue slice, which were further imaged by CLSM to visualize drug distribution in the skin.

### 2.9. Cell Cytotoxicity and Skin Recovery and Irritation Study

The in vitro cytotoxicity of materials used to prepare P@RHT-GMN was performed on HaCaT (human keratinocytes) cells using CCK-8 kit (Dojindo Laboratories, Kumamoto, Japan). The samples were dissolved in PBS and filtered through a 0.22 µm filter. Experiments were carried out in quadruplicate, and cells without any treatment served as the control. HaCaT cells were exposed to dissolved matrix, and the absorbances at 450 nm were measured using a microplate reader following 24 h incubation. The cell viability was determined using the following Equation (2):(2)$$Cell\;viablility\left(\%\right)=\left(A_{sample}-A_{blank}\right)/\left(A_{control}-A_{blank}\right)\times 100\%\;$$
where *A* sample represents the absorbance of a well with cells, CCK-8 reagent, and materials used to prepare P@RHT-GMN; A blank represents the absorbance of a well with medium and CCK-8 reagent without cells; and A control represents the absorbance of a well with cells and CCK-8 reagent without materials used to prepare P@RHT-GMN.

To evaluate the biocompatibility of gas-propelled MNs, an in vivo skin recovery and irritation study was conducted [30]. Rats were anesthetized and the hair on the dorsal skin was shaved. Gas-propelled MNs and passive MNs were applied onto the rat dorsal skin for 3 min. For histopathological analysis, the skin tissues were harvested after gas-propelled MN and passive MN treatment for 24 h, fixed in 10% paraformaldehyde, and then embedded in paraffin. Finally, skin sections were stained with hematoxylin and eosin (H&E) and observed by light microscopy.

### 2.10. Statistical Analysis

All the data were reported as mean ± SD. GraphPad Prism 7.0 software (Graph Pad Software, La Jolla, CA, USA) was used to analyze the data through one-way analysis of variance (ANOVA) among multiple groups or Student’s *t*-test between two groups. The value of *p* < 0.05 was considered statistically significant.

## 3. Results and Discussion

### 3.1. Influence of Processing Technology on the Quality of Male Mode

The state-of-the-art mold processing technology is closely related to the morphology and geometric parameters of MN male mold. However, there are still a lack of systematic studies on how the processing technology affects the quality of MN mold. In this study, four processing technologies, including MEMS-CNC [19], 3D-PuSL [10,24], UV laser drilling [5], and etching [28], were adopted to prepare the male mold of MNs based on the computer-aided precise design of 3D drawings (Figure 2(A1–D1)). The resultant physical images (Figure 2(A2–D2)) showed that all processing technologies can produce the male mold with intact appearance and no broken or missing needles. Next, the longitudinal section of the master structures (Figure 2(A3–D3)) was observed under the visual field of a 4.5 × 50 times microscope. It was found that the needle tips of the brass mold had a certain degree of bending, as the physical processing stress of MEMS-CNC may distort the needle tips where the tip angle is small. The cross-sectional view of the male mold produced by UV laser drilling showed that the needle height differed greatly from each other, thus probably affecting the quality of the MNs in the later stage. This result may be due to the UV laser drilling method having high requirements for the flatness of the silica gel plate and the uniformity of the material. In contrast, the male mold prepared by 3D-PusL and etching was highly uniform and in a good shape.

The geometric parameters of male molds were measured to evaluate the preparation accuracy of different processing technologies. As displayed in Table 5, the male mold prepared by 3D-PusL showed an average needle height (H) of 798 μm, a conical angle (CA) of 24.68°, and a conical base width (CBW) of 360 μm, which were approaching to those of the theoretically designed mold best and exerted minimum batch differences. However, the geometric parameters of male molds prepared by MEMS-CNC, UV laser drilling, and etching were far from the theoretical values, particularly the H and CBW. Moreover, the RSD value obtained from the male molds made from etching was the largest, suggesting that the etching method could produce male molds with large batch differences and low reproducibility. Overall, the 3D-PusL processing technology is the best one to produce male mold with high precision and reproducibility. As observed from the SEM images of the MNs (Figure 2(A4–D4)), the mold processing technology exerted a significant influence on the fineness of the surface thread texture of the MNs, which was consistent with the longitudinal section morphology of the female mold.

### 3.2. Influence of Silica Type on the Quality of Female Mold

Maintaining the morphological integrity of needle tips is the key to successful skin insertion and efficient transdermal drug delivery. Silica gel is a common material used for preparing female mold, and the Shore hardness of silica gel could affect the stress of MNs endured during the demolding process, which will further influence the *DNP* of MNs. Therefore, the influence of silica gel with different Shore hardness on the *DNP* of MNs was investigated. As shown in Figure 3A, the *DNP* of MNs exceeded 85% when the Shore hardness of silica gel was less than 50, and decreased to 79% when it was higher than 50. In general, the smaller the Shore hardness, the larger the *DNP*. This result may be ascribed to the lower stress imposed on the needles by the silica gel with smaller Shore hardness. The longitudinal sections of the female molds prepared from different silica gels (Figure 3B) were observed using a microscope, and showed no noticeable difference in the inner wall of the micropores, except that the female mold made from SP-6010 silica gel was deformed during the longitudinal cutting process. Accordingly, 184 silica gel was suitable for preparing the female mold of MNs. The SEM images of MNs (Figure 3C) showed different fractures of MNs prepared from different negative mold materials. Among these materials, type 184 silicon gel was selected as the optimal material to prepare female mold with minimum needle fractures.

### 3.3. Influence of Microperfusion Technogy on the Formability of MNs

Vacuum micromolding [20,21] and centrifugation micromolding [31] are two commonly used approaches for the preparation of dissolving MNs, in which a highly viscous needle solution is infused into the female hole by pressure gradience and centrifugal force, respectively. Therefore, the influences of different centrifugal speed and negative pressure on the *DNP* and morphology of MNs were investigated. Respective of centrifugation speed, the *DNP* of MNs was less than 40% when the microfilling of the needle suspension was performed by centrifugation (Figure 4A). When the centrifugation speed increased, the sedimentation of the needle suspension could be facilitated to make more solute accumulate in the needle tips, thus reducing the brittleness of the MNs. In contrast, the *DNP* of MNs was increased at higher negative pressure and could exceed 90% when the negative pressure reached −0.08 Mpa and −0.09 Mpa (Figure 4B). Consistent with the result of *DNP*, the MNs prepared by centrifugation micromolding showed large quantities of missing and broken needles as observed from the microscopic images (Figure 4C), while almost no needle fractures or losses were observed in the MNs prepared by vacuum microperfusion at a pressure of −0.08 Mpa and −0.09 Mpa (Figure 4D). The major reason could be ascribed to the rapid sedimentation of particles into the cavity of mold driven by the centrifugation force, which would further cause the discrepancy of particles from the adhesive polymers and dramatic needle breakage at the tip. In contrast, the vacuum micromolding process was mild, which caused no external sedimentation of particles and advantaged to prepare MNs with intact morphology.

### 3.4. Influence of Needle Skeleton Materials on the Formability of MNs

Polymer materials play a critical role in the formability of MNs, as their viscosity and mechanical properties were closely related to the microfilling and demolding process [26]. Therefore, we investigated the influence of several classical polymers on the *DNP* and morphology of MNs. As shown in Figure 5A, the *DNP* of MNs prepared from HPC, PVP K12, PVP K30, and PVP K45 were 98.33%, 22.5%, 98.06%, and 91.11%, respectively. Massive needle shortage and brittleness was obviously observed in PVP K12 MNs (Figure 5(B2)), which may be due to the relatively low viscosity and brittle nature of PVP K12. During the microfilling process, anhydrous ethanol could be accelerated to volatilize at the negative pressure and dramatically increase the viscosity of needle suspension to stop them entering the micropores of mold. Accordingly, the *DNP* of PVP K45 MNs was lower than that of PVP K30 MNs, probably owing to the higher viscosity of the needle suspension imparted by the increased molecular weight. The *DNP* of HPC MNs was comparable to that of PVP K30 MNs. However, the mechanical strength of HPC MNs is usually insufficient for skin penetration. Therefore, PVP K30 was selected as the needle skeleton material of MNs in the further study.

### 3.5. Influence of Drug-Loaded Carriers on the Formability of MNs

The interaction between drug-loaded particles and polymer materials could affect the *DNP* of MNs, and we further optimized the matrix material of drug particles. As demonstrated in Figure 6A, the *DNP* of HA@RHT MNs, DEX@RHT MNs, and PVA@RHT MNs was 60.83%, 88.61%, and 93.06%, respectively. Obvious needle breakage and massive lost needle groups were observed in HA@RHT MNs, but were scarcely observed in EDX@RHT MNs and PVA@RHT MNs (Figure 6B). The drug-loaded suspensions prepared with three materials (HA, EDX, PVA) were further imaged by optical microscopy. The aggregation of HA@RHT particles (Figure 6(C1)) was observed in the needle suspension, which could hinder the needle suspension from entering the micropores of female mold smoothly, and further cause massive needle deficiency. As the lowest aggregation and highest *DNP* were obtained in the presence of PVA@RHT particles, PVA was selected as the drug carrier to prepare particle-embedded gas-propelled MNs. The optimal feeding concentration of PVA@RHT particles was further screened by characterizing the *DNP* and morphology of MNs. The results demonstrated that the MNs could achieve a *DNP* exceeding 92% (Figure 7A) and maintain the integrity of the needle (Figure 7B) at a feeding concentration of PVA@RHT particles less than 0.2 g/mL.

### 3.6. Influence of Pneumatic Initiators on the Formability of MNs

A certain amount of alkaline agent (K_2_CO_3_ and Na_2_CO_3_) and acid agent (CA and TA) were introduced in the needle suspension as the pneumatic initiators to prepare gas-powered MNs. We first investigated the compositions of pneumatic initiators on the effects of *DNP* and the morphology of MNs. Regardless of using CA or TA as the acid agent, their combination with K_2_CO_3_ produced a *DNP* above 90%, which was higher than that of Na_2_CO_3_ (about 80%). Moreover, the combination of K_2_CO_3_ with CA achieved the highest *DNP* of about 93.61% (Figure 8A). The macroscopic images (Figure 8B) also showed that the gas-propelled MNs showed dramatic needle deficiency and broken needles in the presence of Na_2_CO_3_, especially when using Na_2_CO_3_ and CA as the pneumatic initiators. As K_2_CO_3_ was added as solid particles in the needle suspension that could affect the formability of MNs, we further investigated how the feeding concentration of K_2_CO_3_ and CA influenced the *DNP* and morphology of MNs. It was found that the higher feeding concentration of K_2_CO_3_ and CA produced a lower *DNP* (Figure 8A). When the feeding concentrations of K_2_CO_3_ and CA were 0225:0.225 (*w*/*w*), 0.15:0.15 (*w*/*w*), 0.075:0.075 (*w*/*w*), and 0.0375:0.0375 (*w*/*w*), the correspondent DNPs of MNs were 79.17%, 93.61%, 90.28%, and 95.28%, respectively. The probable reason was that the interaction force between solid K_2_CO_3_ particles and polymers was weak during preparation, resulting in increased brittleness and fracture during the demolding of MNs. Taking both the *DNP* and gas-generating productivity of MNs into account, K_2_CO_3_ and CA at the proportion of 0.15:0.15 (*w*/*w*) were used as the pneumatic initiators to prepare gas-propelled MNs for further study.

### 3.7. Morphology, Drug Loading, and Drug Distribution of MNs

As displayed in Figure 9A, the optimized gas-propelled MNs showed a cone needle shape without cracks, which was almost the same as those of passive MNs and free RHT-loaded MNs. It was indicated that the introduction of either drug particles or pneumatic initiators did not alter the morphology of MNs after optimizing the formulation components and preparation process. Compared to the free RHT-loaded MNs, the drug loading of MNs laden with RHT particles increased from 503.9 μg to 683.37 μg per patch (Figure 9B), which was increased by 35%. This result agreed well with the previous reports that showed that replacing the free drug with concentrated drug particles served as an effective approach to increase drug loading [10]. Furthermore, RhB was used to prepare fluorescence-labeled MNs and denote the drug distribution in MNs. As observed by the CLSM, the 3D reconstruction of passive and gas-propelled MNs (Figure 9C) revealed that the fluorescence signal was detected in the whole needle, and pneumatic initiators exerted almost no influence on drug distribution. 

### 3.8. In Vitro and In Vivo Transdermal Permeability of MNs

The Franz diffusion cell was further used to evaluate the transdermal permeability of MNs. As expected, the permeation of RHT from gas-propelled MNs was significantly faster than that from conventional passive MNs, resulting in cumulative permeability approaching 100% at 36 h (Figure 9D). Compared to the passive MNs (79.67%), the cumulative permeability of gas-propelled MNs showed a 1.19-fold increase. RhB was used to replace RHT for the preparation of fluorescence-labeled MNs that were further imaged by CLSM to observe the drug permeation in the excised rat skin. It was found that the gas-propelled MNs could promote the permeation of RhB into deeper skin layers at a larger area than the passive MNs (Figure 9E). The RhB-labeled MNs were further applied to the living rat skin for confirming whether the gas-propelled MNs could effectively promote drug permeation in vivo. Consistent with the in vitro results, stronger fluorescence intensity could be observed in the skin tissues ranging from the epidermis to the dermis at the same timepoint (Figure 9F), suggesting that a larger amount of the drug could be delivered to deeper skin more rapidly. These results collectively confirm the feasibility of gas-propelled MNs to promote drug permeation and enhance transdermal delivery efficiency, as the CO_2_ bubbles produced by the chemical reaction of K_2_CO_3_ and CA could exert an intensive propulsion force to facilitate drug diffusion and permeation into deeper skin tissues [32,33].

### 3.9. Cell Cytotoxicity and Skin Irritation Study

A cytotoxicity study was used to evaluate the biosafety of the materials used to manufacture the gas-propelled MNs. As shown in Figure 10A, there was no significant difference between the passive MNs and gas-propelled MNs in terms of cell viability after 24 h of incubation, both being above 90%, indicating that the materials used in this study were nontoxic.

Histological studies of the administration site were performed to further confirm the biocompatibility of the gas-propelled MNs. As shown in Figure 10B, the gas-propelled MN group did not produce a proinflammatory response in the skin, indicating that the gas bubbles produced by gas-propelled MNs do not cause any adverse reaction.

## 4. Conclusions

This work provides a systematic study on the preparation of microparticle-embedded gas-propelled MNs from the very beginning to the end. The processing technologies could exert critical influence on the geometric parameters and fineness degree of male mold. The 3D PuSL works best to produce male mold with high precision and reproducibility, as evidenced with uniform geometric parameters that are most approachable to the theoretical designed values. Notably, the Shore hardness of silica gel was found to be inversely correlated with the *DNP* of MNs. The gas-propelled MNs were further optimized by studying the effect of micromolding technologies, needle skeleton polymers, drug particle carriers, and pneumatic initiators on the *DNP* and morphologies of MNs. The *DNP* of gas-propelled MNs prepared by vacuum micromolding under optimal negative pressure could exceed 90%, which was far greater than that of MNs prepared by centrifugation micromolding. The formulation optimization studies demonstrated that the gas-propelled MNs could obtain the highest *DNP* and intact needles when using PVP K30, PVA, and K_2_CO_3_:CA = 0.15:0.15 (*w*/*w*) as the needle skeleton material, drug particle carrier, and pneumatic initiators, respectively. The feasibility of gas-propelled MNs to simultaneously enhance the drug loading and transdermal delivery efficiency in vitro and in vivo was also confirmed. It should also be noted that a major challenge of gas-propelled MNs is to prevent the chemical reaction between K_2_CO_3_ and CA during preparation and storage, since the gas-generating ability of MNs is critical to produce the propulsion force for promoting drug permeation in the skin. Therefore, the preparation of gas-propelled MNs is usually conducted in an anhydrous environment, and the humidity of the environment needs to be strictly controlled during storage. Overall, this study provides detailed guidance for preparing gas-propelled MNs with excellent productivity, high drug loading, and improved delivery efficiency that show great potential for clinical applications.

## Figures and Tables

**Figure 1 pharmaceutics-15-01059-f001:**
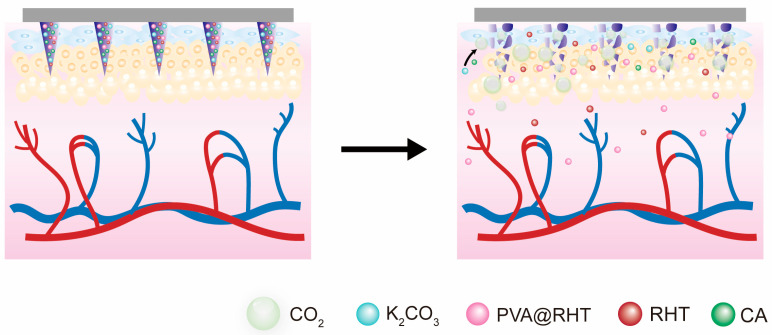
The mechanism diagram of the pneumatic initiators.

**Figure 2 pharmaceutics-15-01059-f002:**
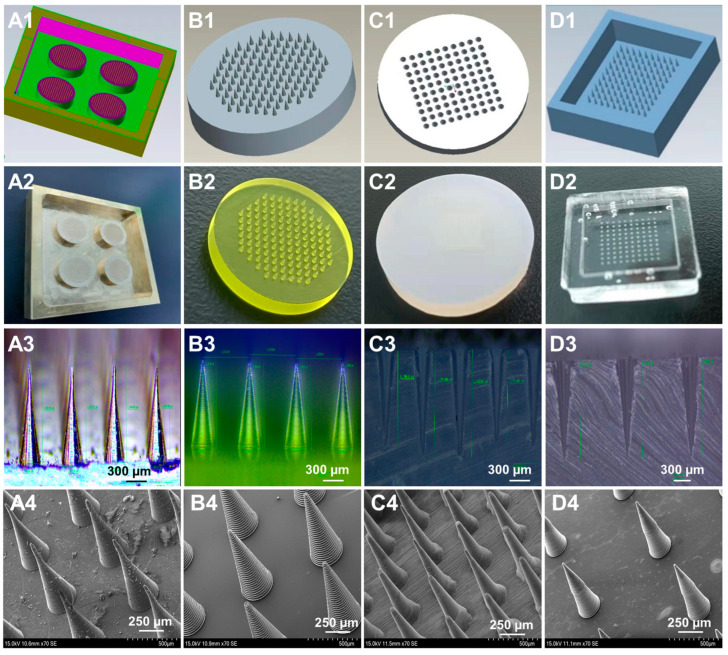
The 3D design drawings (**A1**,**B1**,**C1**,**D1**), physical pictures (**A2**,**B2**,**C2**,**D2**), and longitudinal sections (**A3**,**B3**,**C3**,**D3**) of male mold prepared by different processing technologies; the SEM images of MNs prepared by different mold processing technologies (**A4**,**B4**,**C4**,**D4**): (**A**) MEMS-CNC, (**B**) 3D-PuSL, (**C**) UV laser drilling, and (**D**) etching.

**Figure 3 pharmaceutics-15-01059-f003:**
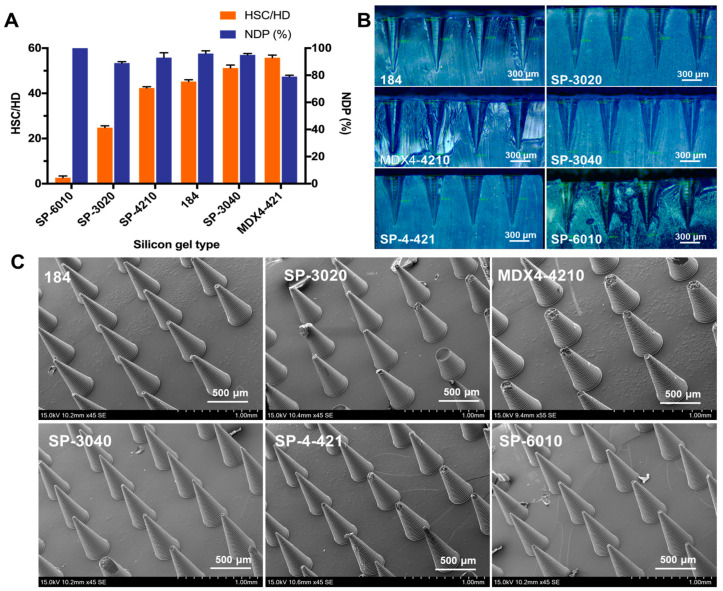
(**A**) The correlation between the Shore hardness of silica gel and the *DNP* of MNs (*n* = 3). (**B**) The cross-sectional view of female molds prepared from different types of silica gel. (**C**) The SEM images of MNs prepared by different female mold preparation material.

**Figure 4 pharmaceutics-15-01059-f004:**
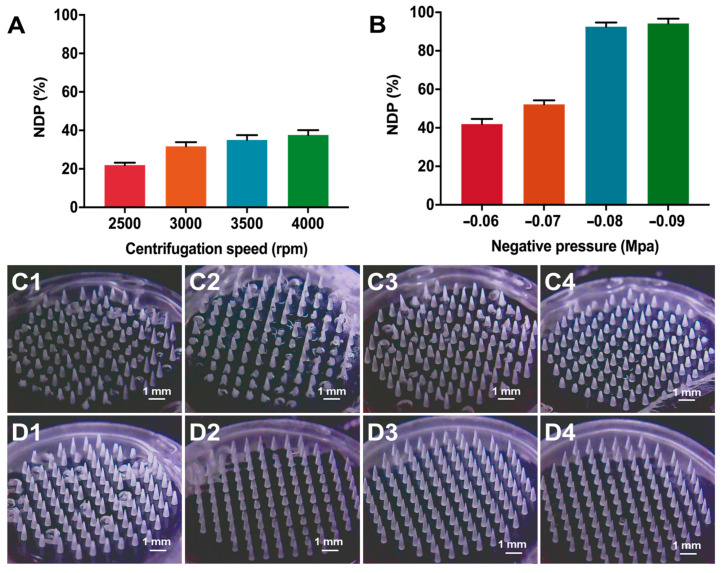
The influence of (**A**) centrifugation speed and (**B**) negative pressure on the *DNP* of particle-embedded gas-propelled MNs (*n* = 3). The morphology of MNs prepared at the speed of (**C1**) 2500 rpm, (**C2**) 3000 rpm, (**C3**) 3500 rpm, (**C4**) 4000 rpm, or at the negative pressure of (**D1**) −0.06 Mpa, (**D2**) −0.07 Mpa, (**D3**) −0.08 Mpa, (**D4**) −0.09 Mpa.

**Figure 5 pharmaceutics-15-01059-f005:**
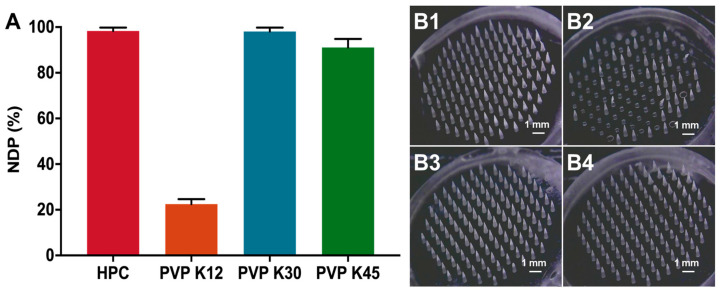
The influence of polymers used as needle skeleton materials on the (**A**) *DNP* (*n* = 3) and (**B**) morphology of particle-embedded gas-propelled MNs: (**B1**) HPC, (**B2**) PVP K12, (**B3**) PVP K30, and (**B4**) PVP K45.

**Figure 6 pharmaceutics-15-01059-f006:**
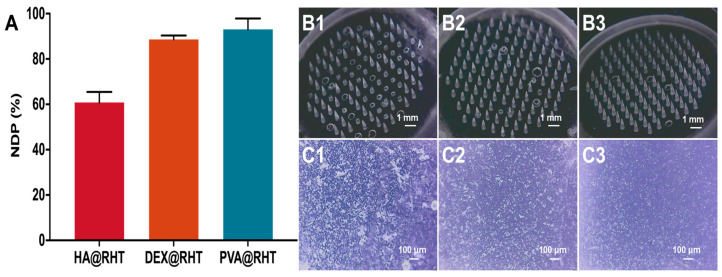
The influence of drug-loaded carriers on the (**A**) *DNP* (*n* = 3) and (**B**) morphology of particle-embedded gas-propelled MNs: (**B1**) HA@RHT, (**B2**) DEX@RHT, and (**B3**) PVP@RHT. (**C**) The influence of carriers on the morphology of drug particles: (**C1**) HA@ RHT, (**C2**) DEX@RHT, and (**C3**) PVP@RHT.

**Figure 7 pharmaceutics-15-01059-f007:**
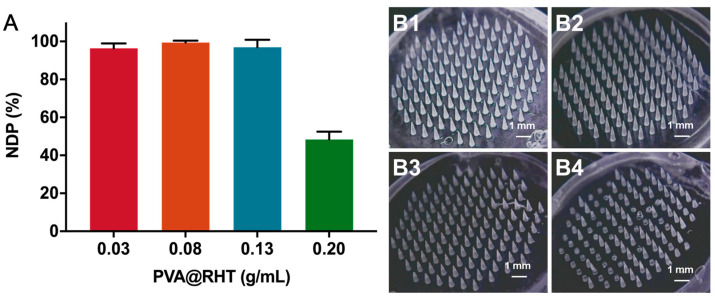
The influence of PVP@RHT feeding concentration on the (**A**) *DNP* (*n* = 3) and (**B**) morphology of particle-embedded gas-propelled MNs: (**B1**) 0.03 g/mL, (**B2**) 0.08 g/mL, (**B3**) 0.13 g/mL, and (**B4**) 0.20 g/mL.

**Figure 8 pharmaceutics-15-01059-f008:**
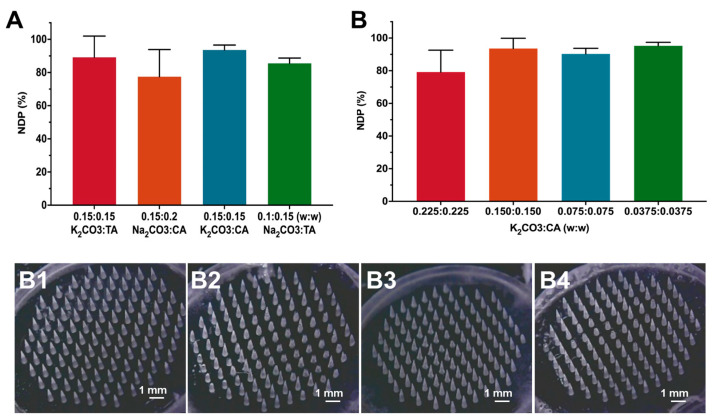
The influence of pneumatic initiator compositions on the (**A**) *DNP* (*n* = 3) and (**B**) morphology of particle-embedded gas-propelled MNs: (**B1**) K_2_CO_3_:TA = 0.15:0.15, (**B2**) Na_2_CO_3_:CA = 0.15:0.20, (**B3**) K_2_CO_3_:CA = 0.15:0.15, (**B4**) Na_2_CO_3_:TA = 0.1:0.15. (**B**) The influence of the feeding concentration of K_2_CO_3_ and CA on the *DNP* of MNs (*n* = 3).

**Figure 9 pharmaceutics-15-01059-f009:**
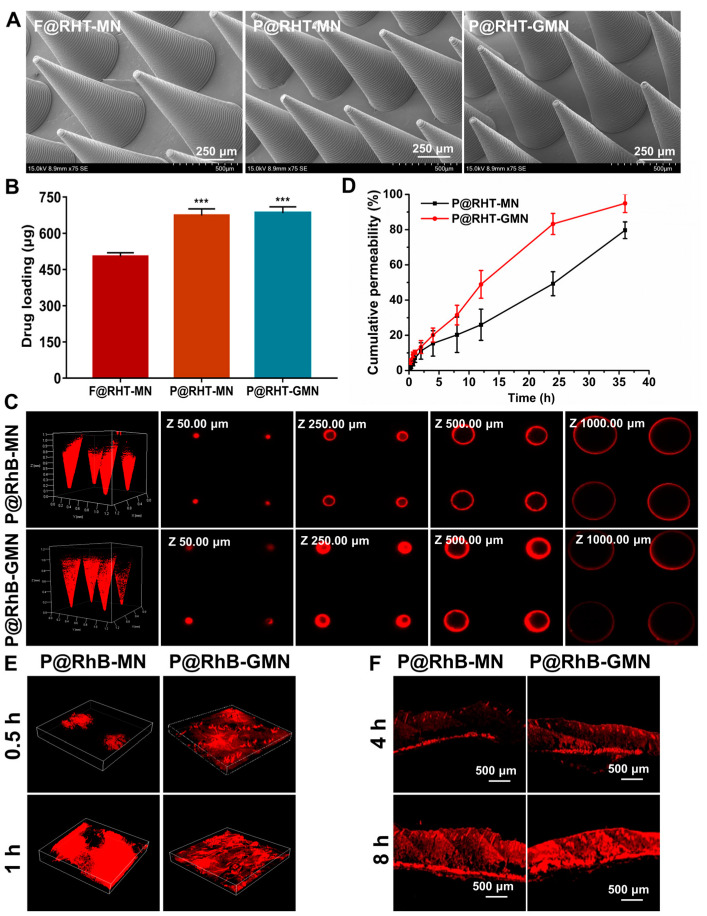
Characterization of optimized MNs: (**A**) morphology of MNs observed by SEM; (**B**) drug loading (*n* = 3). *** *p* < 0.001 vs. F@RHT-MN; (**C**) the 3D reconstruction of P@RHT-MN and P@RHT-GMN and at their counterparts at different depths captured by CLSM to observe drug distribution; (**D**) in vitro cumulative transdermal permeability of RHT (*n* = 3); the permeation of RhB into the (**E**) excised and (**F**) living skin tissues to evaluate the transdermal delivery efficiency of passive and gas-propelled MNs.

**Figure 10 pharmaceutics-15-01059-f010:**
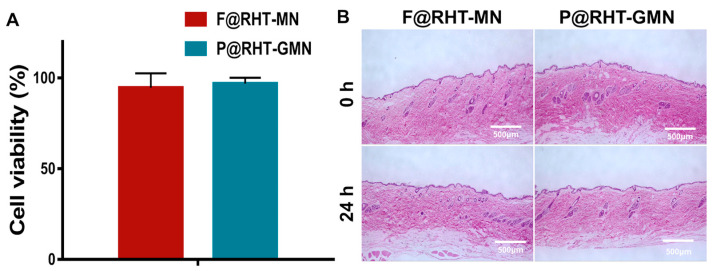
(**A**) The cytotoxicity of passive MNs and gas-propelled MNs after incubating HaCaT cells for 24 h (*n* = 5). (**B**) The skin irritation of passive MNs and gas-propelled MNs.

**Table 1 pharmaceutics-15-01059-t001:** Compositions of different materials for preparing male mold.

Name	Type	Silica Gel:Curing Agent (*w*/*w*)
PDMS	184	10:1
Silicone elastomers	MDX4-4210	10:1
Liquid silica gel-1	SP-6010	1:1
Liquid silica gel-2	SP-3020	1:1
Liquid silica gel-3	SP-3040	1:1
Liquid silica gel-4	SP-4210	1:1

**Table 2 pharmaceutics-15-01059-t002:** Types and feeding concentrations of polymers for preparing needles.

Formulations	Polymers	Feeding Concentration (g/mL)
F1	HPC	0.16
F2	PVPK12	0.47
F3	PVPK30	0.47
F4	PVPK45	0.26

**Table 3 pharmaceutics-15-01059-t003:** Formulation compositions and feeding concentrations of drug particles.

Abbreviation	Drug-Loaded Particles	Feeding Concentration (g/mL)
P1	HA@RHT	0.07
P2	DEX@RHT	0.07
P3	PVA@RHT	0.07
P4	PVA@RHT	0.03
P5	PVA@RHT	0.08
P6	PVA@RHT	0.13
P7	PVA@RHT	0.20

**Table 4 pharmaceutics-15-01059-t004:** Optimization of pneumatic initiators.

No.	Pneumatic Initiators	Feeding Concentration (*w*/*w*)	Mol:Mol
S1	K_2_CO_3_:TA	0.15:0.15	1:1
S2	Na_2_CO_3_:CA	0.15:0.20	3:2
S3	K_2_CO_3_:CA	0.15:0.15	3:2
S4	Na_2_CO_3_:TA	0.10:0.15	1:1
S5	K_2_CO_3_:CA	0.0375:0.0375	3:2
S6	K_2_CO_3_:CA	0.0075:0.0075	3:2
S7	K_2_CO_3_:CA	0.15:0.15	3:2

**Table 5 pharmaceutics-15-01059-t005:** Geometric parameters of male mold prepared by different methods (*n* = 6).

Method	Index	Geometric Parameters						Mean ± SD	RSD%
MEMS-CNC	H ^a^ (μm)	841	853	844	829	836	838	840 ± 8	0.96
	CA ^b^ (°)	22.29	22.89	22.34	23.50	23.03	22.79	22.81 ± 0.45	1.98
	CBW ^c^ (μm)	353	344	344	357	352	348	350 ± 5	1
3D-PuSL	H ^a^ (μm)	793	809	800	782	805	798	798 ± 10	1.20
	CA ^b^ (°)	24.48	24.83	24.86	24.87	24.69	24.37	24.68 ± 0.21	0.86
	CBW ^c^ (μm)	364	362	355	360	356	365	360 ± 4	1.00
UV laser drilling	H ^a^ (μm)	731	814	834	709	729	722	756 ± 53	3
	CA ^b^ (°)	16.44	16.56	14.36	17.05	14.83	18.58	16.30 ± 1.53	5.41
	CBW ^c^ (μm)	244	216	240	229	251	197	230 ± 20	5
Etching	H ^a^ (μm)	711	713	713	670	675	668	692 ± 23	7
	CA ^b^ (°)	24.26	23.55	22.18	21.83	22.19	20.90	22.49 ± 1.22	9.41
	CBW ^c^ (μm)	279	277	283	314	296	283	289 ± 14	9

^a^ H refers to the needle height. ^b^ CA refers to the conical angle of needles. ^c^ CBW refers to the conical base width of needles.

## Data Availability

Not applicable.

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
