# Peer review of "State of the Art in Constructing Gas-Propelled Dissolving Microneedles for Significantly Enhanced Drug-Loading and Delivery Efficiency"

_pharmaceutics, 2023, doi:10.3390/pharmaceutics15041059_

Round 1

Reviewer 1 Report

The authors performed a comprehensive examination of the entire process of creating gas-propelled, microparticle-embedded microneedles. I believe that the study will be of interest to potential leaders, however there are improvements that could be made to the manuscript to improve its impact/benefit to readers.

Line 129: Quite a key formula seems to be missing, that has many references throughout the paper. A re-review will be necessary simply because of this error.

Line 160, and others: Correct, subscripted forms of chemical formulae should be used.

Line 365: As there is extensive talk about microneedle forming failure, and since I perceive this to be one of the main points made in the manuscript, I believe more detailed images (than the macroscopic images provided) is necessary. SEM images would be ideal to look at the surfaces/cracks/deformations of the microneedles in detail. The exact story and argument made is difficult to get from the images provided.

Author Response

The authors performed a comprehensive examination of the entire process of creating gas-propelled, microparticle-embedded microneedles. I believe that the study will be of interest to potential leaders, however there are improvements that could be made to the manuscript to improve its impact/benefit to readers.

Line 129: Quite a key formula seems to be missing, that has many references throughout the paper. A re-review will be necessary simply because of this error.

Response: The formula to calculate demolding needle percentage (DNP) has been supplemented in the revised manuscript (Page 4, Line 167) and shown as the follows.

Line 160, and others: Correct, subscripted forms of chemical formulae should be used.

Response: All the chemical formulae have been corrected and presented in the subscripted forms in the revised manuscript (Page 6, Lines 234-235, Page 7, Line 241).

Line 365: As there is extensive talk about microneedle forming failure, and since I perceive this to be one of the main points made in the manuscript, I believe more detailed images (than the macroscopic images provided) is necessary. SEM images would be ideal to look at the surfaces/cracks/deformations of the microneedles in detail. The exact story and argument made is difficult to get from the images provided.

Response: Thanks for your kind advice. Based on our previous study, the microstructure of microneedles regrading to the surfaces/cracks/deformations was mainly affected by the male mold processing technology and female mold preparation material. Therefore, the SEM images (Figure 2A4/B4/C4/D4 and Figure 3C) of these microneedles were supplemented in the revised manuscript (Page 9, Lines 327-334 and Page 10, Lines 354-357) and shown as the following. The results showed that the mold processing technology could exert appreciable effects on the fineness of surface threaded texture of microneedles and the microneedle with clear threaded texture could be prepared by the 3D-PusL processing technology, while the preparation material of male mold mainly affected the cracks of microneedles. Besides, the SEM images of optimized gas-propelled MNs was also presented in Figure 9A and showed that the MNs were in cone shape without needle cracks and remained the same morphology of passive MNs and free RHT loaded MNs (Page 15, Line 461-465).

Reviewer 2 Report

Brief summary:

Microneedles are considered as promising drug delivery platform. However, the development and clinical studies of microneedle applicators are complicated by various factors. Thus, mass microneedles production requires the creation of standardized methods and specialized automated quality control systems. In this work a detailed guidance for preparing gas-propelled microneedles with excellent productivity, high drug loading, and improved delivery efficiency was suggested by the authors. I have read the manuscript carefully and found it well written, well-structured and the results are presented very nicely. Thus, I can recommend this work for publication in the journal after minor revision.

Major comments:

1) The formula is missing (line 129).

2) Section 2 (Materials and Methods) should be added with a “Statistical analysis” subsection.

3) A detailed description of work with mice (detention conditions, ethical approval, etc.) should be added to the “Materials and Methods” section.

Minor comments:

1) Please specify the manufacturer's country for all software and devices mentioned in the paper as you do it for Shore hardness tester (LX-C, China) (line 143) for example. At the moment there are several positions with lack of this information: i.e. AutoCad (line 108), SolidWorks (line 109) etc.

2) Indexes in formulas must be subscript. At the moment, in the article, some formulas are written correctly (for example, lines 359, 363, 374), and some are not (K2CO3 in line 160, K2CO3 and Na2CO3 in line 199, CO2 in line 200 and others).

3) “Materials and Methods” section should contain a detailed description of all the operations used in this study. At the same time, it should not contain information related to the selection or discussion of these operations. Thus, in my opinion, the text “Vacuum micromolding and centrifugation micromolding are two commonly used approaches for preparing dissolving MNs, while their influence on the DNP and morphology of gas-powered MNs remained unknown” (lines 177-179) should be transferred to the “Results and Discussion” section.

4) “In vivo” and “in vitro” should be in cursive (lines 218, 413, 416 and 435).

5) “Conclusions” section should be numbered as 4, not as 5.

Author Response

Microneedles are considered as promising drug delivery platform. However, the development and clinical studies of microneedle applicators are complicated by various factors. Thus, mass microneedles production requires the creation of standardized methods and specialized automated quality control systems. In this work a detailed guidance for preparing gas-propelled microneedles with excellent productivity, high drug loading, and improved delivery efficiency was suggested by the authors. I have read the manuscript carefully and found it well written, well-structured and the results are presented very nicely. Thus, I can recommend this work for publication in the journal after minor revision.

Major comments:

1) The formula is missing (line 129).

Response: The formula has been added in the revised manuscript (Page 4, Line 167).

2) Section 2 (Materials and Methods) should be added with a “Statistical analysis” subsection.

Response: The “Statistical analysis” subsection has been added the revised manuscript (Page 8, Lines 293-297) and shown as the following.

All the data was reported as mean ± SD. The GraphPad Prism 7.0 software (Graph Pad Software, La Jolla, CA) was used to analyze the data through one-way analysis of variance (ANOVA) among multiple groups or student’s t-test between two groups. The value of P < 0.05 was considered statistically significant.

The statistical analysis has been added to Figure 9B and 9D in the revised manuscript.

3) A detailed description of work with mice (detention conditions, ethical approval, etc.) should be added to the “Materials and Methods” section.

Response: Thanks for your kind advice, the mice information has been added to the “Materials and Methods” section (Page 3, Lines 111-118) and shown as follows:

The healthy male SD rats (220-250 g) and male C57BL/C female mice (16-20 g) were purchased from the Experimental Animal Center of Southern Medical University (License number: SCXK (Guangdong) 2021-0041). The animal experiment has been reviewed by the Experimental Animal Ethics Committee of Sun Yat-sen University and complies with the relevant regulations of the National Experimental Animal Welfare Ethics. The approval number was SYSU-IACUC-2022-001871. The rats were raised in an SPF environment with a temperature of 25 ± 2 °C, a relative humidity of 40-70 %, and an illumination of 15-20 lx.

Minor comments:

1) Please specify the manufacturer's country for all software and devices mentioned in the paper as you do it for Shore hardness tester (LX-C, China) (line 143) for example. At the moment there are several positions with lack of this information: i.e. AutoCad (line 108), SolidWorks (line 109) etc.

Response: The country has been added to all the software and devices mentioned above in the revised manuscript (Page3, Lines 120 and 122).

2) Indexes in formulas must be subscript. At the moment, in the article, some formulas are written correctly (for example, lines 359, 363, 374), and some are not (K2CO3 in line 160, K2CO3 and Na2CO3 in line 199, CO2 in line 200 and others).

Response: All the indexes in formulas have been corrected as subscript form in the revised manuscript (Page 6, Lines 234-235, Page 7, Line 241).

3) “Materials and Methods” section should contain a detailed description of all the operations used in this study. At the same time, it should not contain information related to the selection or discussion of these operations. Thus, in my opinion, the text “Vacuum micromolding and centrifugation micromolding are two commonly used approaches for preparing dissolving MNs, while their influence on the DNP and morphology of gas-powered MNs remained unknown” (lines 177-179) should be transferred to the “Results and Discussion” section.

Response: Thanks for your kind advice, a detailed description of the operation methods has been added in the revised manuscript, including the preparation parameters of the female mold and the principle of the MEMS-CNC, 3D PuSL, UV laser drilling and chemical etching (Page 3-4, Lines 131-155).

Also, the text “Vacuum micromolding and centrifugation micromolding are two commonly used approaches for preparing dissolving MNs, while their influence on the DNP and morphology of gas-powered MNs remained unknown” has been transferred to the “Results and Discussion” section (Page 11, Line 363-364).

4) “In vivo” and “in vitro” should be in cursive (lines 218, 413, 416 and 435).

Response: “In vivo” and “in vitro” have presented in cursive in the revised manuscript (Page 15, Line 474/ 485).

5) “Conclusions” section should be numbered as 4, not as 5.

Response: This error has been corrected in the revised manuscript (Page 17, Line 513)

Reviewer 3 Report

The manuscript reports microparticle embedded gas-propelled microneedles for simultaneously enhancing drug loading and transdermal delivery efficiency. The design of the experiments and results are properly reported, however there are several issues that need to be improved and expanded.

1- Avoid using abbreviations before defining them in the abstract. Write the full names of for PVP K30, PVA, K2CO3:CA in the abstract.

2- Selection of PVP K30, PVA, K2CO3:CA for the microneedle materials has been not discussed adequately. Explain and give reasoning for the selection of these specific materials.

3- There are many writing and grammatical errors some of which are as follows:

- Line 21: "micrmolding"

- Line 73: "mircoholes"

- Line 167, "mirofilling"

- Line 212, "Rhodamine were"

- Line 211, add space between numerical value and the unit: 1mL/min

4- Line 58: The effect of the specified factors on the microneedle should be explained because that is the main goal of the article.

5- Line 129, DNP calculation formula is missing.

6- Line 142, what is NDP? This abbreviation has not been defined before but used for many times in the following pages.

7- Figure 3C1-C4, it seems the needle breakage reduced by increasing centrifugation speed. Please discuss why.

8- Line 300, the pressure value of 0.08 MPa seems to be wrong. It should be negative. Also, the units for pressure values should be MPa, instead of Mpa.

9- Figure 1 and 2B, 3C, 3D, 5B, 5C and others where necessary: Scale bars should be added to the images.

10- The techniques for the male mold preparation should be discussed in more detail where further experimental parameters need to be added. MEMS-CNC, 3D-PuSL, UV laser drilling, and etching are mentioned without much detail on their operation principles.

11- The limitations of challenges of gas-propelled microneedles should be provided along with possible solutions. Are there any side effects of the generated carbon dioxide? Will they induce toxicity etc.

Author Response

The manuscript reports microparticle embedded gas-propelled microneedles for simultaneously enhancing drug loading and transdermal delivery efficiency. The design of the experiments and results are properly reported, however there are several issues that need to be improved and expanded.

1- Avoid using abbreviations before defining them in the abstract. Write the full names of for PVP K30, PVA, K2CO3:CA in the abstract.

Response: The full names of PVP K30, PVA, K2CO3 and CA in the abstract have been added to the revised manuscript (Page 1, Line 27-28) and were shown as the follows.

Polyvinyl Alcohol (PVA); Polyvinylpyrrolidone K30 (PVP K30); Potassium carbonate (K2CO3); Citric Acid (CA)

2- Selection of PVP K30, PVA, K2CO3:CA for the microneedle materials has been not discussed adequately. Explain and give reasoning for the selection of these specific materials.

Response: The introduction of K2CO3 and CA in microneedles was to produce carbon dioxide (CO2) bubbles for promoting drug permeation in the skin. To avoid the chemical reaction between K2CO3 and CA, the preparation of gas-propelled microneedles is usually carried out under anhydrous conditions, and therefore PVP K30 and PVA that could dissolve in ethanol were selected as the needle matrix material. Besides, the combination of PVP K30 and PVA was reported to improve the mechanical strength of MNs [1].

Reference

[1] TPGS/hyaluronic acid dual-functionalized PLGA nanoparticles delivered through dissolving microneedles for markedly improved chemo-photothermal combined therapy of superficial tumor. Acta Pharm Sin B 2021, 11, 3297-3309, doi:10.1016/j.apsb.2020.11.013.

3- There are many writing and grammatical errors some of which are as follows:

- Line 21: "micrmolding"

- Line 73: "mircoholes"

- Line 167, "mirofilling"

- Line 212, "Rhodamine were"

- Line 211, add space between numerical value and the unit: 1mL/min

Response: These errors have been corrected in the revised manuscript.

"micromolding" -(Page 1, Line 21)

"microholes"-(Page 2, Line 76)

"microfilling"-(Page 5, Line 204)

"rhodamine B was"-(Page 7, Line 253)

Add space between numerical value and the unit: 1mL/min-(Page 7, Line 250)

4- Line 58: The effect of the specified factors on the microneedle should be explained because that is the main goal of the article.

Response: The effect of the specific factors has been described in the revised manuscript (Page 2, Lines 59-74) and shown as the following.

A variety of physical and chemical permeation-enhancement techniques have been integrated with MNs to improve drug delivery efficiency. Common physical methods include infrared light, electric field, or ultrasonic irradiation. Pre-exposure of the skin to infrared light, electric fields or ultrasound, followed by application of the MNs on the skin, or a synchronous treatment regimen, can effectively enhance the permeability of the drug[2]. However, infrared light and ultrasonic irradiation require external equipment and professionals to assist drug administration, which are complicated to operate and there is a lack of quantitative studies between relevant parameters and penetration effects. Electric fields are only capable of delivering ionic compounds, and penetration of drug species and transdermal drug delivery remains limited.  

The combination of MNs and chemical permeation enhancers has also been reported to increase the permeability of drug across skin. For example, amphoteric ionophores, degradative enzyme inhibitors, and vasodilators increase the amount of drug penetration by increasing the fluidity of cell membranes and the permeability of tissue interstitial spaces. However, a large amount of chemical permeation enhancers is usually required to dramatically increase the transdermal delivery efficiency, which may lead to reduced MNs mechanical strength, as well as vasodilators that can irritate blood vessels and increase the risk of skin irritation and safety issues.

Reference

[2] Bhatnagar, S.; Kwan, J.J.; Shah, A.R.; Coussios, C.C.; Carlisle, R.C. Exploitation of sub-micron cavitation nuclei to enhance ultrasound-mediated transdermal transport and penetration of vaccines. J Control Release 2016, 238, 22-30, doi:10.1016/j.jconrel.2016.07.016.

5- Line 129, DNP calculation formula is missing.

Response: The formula has been added in the revised manuscript (Page 4, Line 167) and shown as the follows.

Where, NT represents the number of needles with intact morphology counted from the macroscopic images of MN arrays, and NS represents the number of needles based on the original design.

6- Line 142, what is NDP? This abbreviation has not been defined before but used for many times in the following pages.

Response: The NDP that represents the demolding needle percentage has been changed to DNP in the revised manuscript (Page 4, Line 179, Page 11, Line 359/367/368/371).

7- Figure 3C1-C4, it seems the needle breakage reduced by increasing centrifugation speed. Please discuss why.

Response: With the centrifugation speed increased, the sedimentation of needle suspension could be facilitated to make more solute accumulated in the needle tips, thus reducing the brittleness of the MNs. The detailed reason has been added in the revised manuscript (Page 11, Lines 369-371).

8- Line 300, the pressure value of 0.08 MPa seems to be wrong. It should be negative. Also, the units for pressure values should be MPa, instead of Mpa.

Response: This error has been corrected in the revised manuscript (Page 6, Line 214-215, Page 11, Lines 373 and 377, Page 12, Line 387).

9- Figure 1 and 2B, 3C, 3D, 5B, 5C and others where necessary: Scale bars should be added to the images.

Response: Scale bars have been added to the images (Figure 2A3-D4, 3C, 4C, 4D, 5B, 6B, 6C, 7B, 8B) and were shown as the follows.

10- The techniques for the male mold preparation should be discussed in more detail where further experimental parameters need to be added. MEMS-CNC, 3D-PuSL, UV laser drilling, and etching are mentioned without much detail on their operation principles.

Response: Thanks for your kind advice. The experimental parameters for the male mold preparation and the principles of MEMS-CNC, 3D-PuSL, UV laser drilling, and etching have been added in the revised manuscript (Page 3, Line 131-133 and Page 4, Line 134-155) and described as below:

(1) The MEMS-CNC method is to fix the brass material on the high-precision machine tool (JDMR600, Beijing, China) and install the milling cutter with a taper angle of 23° in the spindle tool slot. The program input is made according to the three-dimensional design drawing, and then the milling cutter is rotated at 15000 rpm to mill the conical main mold.

(2) The 3D-PuSL is to inject the high temperature-resistant photosensitive resin material into the resin tank through the pump liquid system of nanoArch®P140 3D printe (nanoArch ® P140, Chongqing, China). The designed 3D model of MNs is imported into the high precision ultraviolet lithography projection system, followed by instantaneous exposure to 405 nm UV light for curing. Then, the male mold of MNs is prepared by layer printing.

(3) The ultraviolet laser drilling is to use a laser (FM-UVM5, Shanghai, China) with a wavelength of 355 nm, 65% output power (adjustable power range: 1-10 W) and 150 mm/s laser speed (adjustable speed range: 100-200 mm/s), a pulse frequency of 30 kHz, to prepare MN mold on the surface of a 5 mm thick silicone plate by one-time burning.

(4) In the etching method, the thick silicon nitride protective film was deposited on both sides of the silicon wafer by the low-pressure chemical vapor deposition (LPCVD) technology, and the photoresist was spun. The circular spot pattern of the mask was transferred to the photoresist to form the blocking adhesive film. Then, dry etching was carried out by inductively coupled plasma etching system. After cleaning up the treated silicon wafer with deionized water, the isotropic wet etching was performed to obtain the MN mold.

11- The limitations of challenges of gas-propelled microneedles should be provided along with possible solutions. Are there any side effects of the generated carbon dioxide? Will they induce toxicity etc.  

Response: The major challenge of gas-propelled MNs is to prevent the chemical reaction between K2CO3 and CA during preparation and storage, since the gas-generating ability of MNs is critical to produce the propulsion force for promoting drug permeation in the skin. Therefore, the preparation of gas-propelled MNs is usually conducted under the anhydrous environment and the humidity of the environment needs to be strictly controlled during storage. These descriptions have been added in the revised manuscript (Page 17, Line 530-535).

The cell cytotoxicity and in vivo skin irritation studies have shown that the carbon dioxide (CO2) produced by gas-propelled MNs is biocompatible and produced no cytotoxicity and skin irritation (Figure 10A-10B). These results have been added in the revised manuscript (Page 16-17, Lines 500-509) and shown as the following.

Reviewer 4 Report

This article State-of-the-art in constructing gas-propelled dissolving mi-croneedles for significantly enhanced drug loading and deliv-ery efficiency introduces optimization of microparticle-embedded gas-propelled microneedles preparation technologies and formulation composition which enhances drug loading and transdermal delivery efficiency. The approach of this manuscript is considered to be an informative one; however, major revision is recommended due to the following points.

- The novelty of the manuscript is the concept of using gas-propelled microneedles which enhance local drug absorption. It is recommended to provide a figure for mechanism of pneumatic initiators which were utilized to produce carbon dioxide (CO2) bubbles through acid-base reaction in water, thus accelerating the dissolution of MNs. There is lack of data to detail the novelty of gas propelled microneedles.

- On page 2 line 86, “Based above knowledge” should be “Based on above knowledge”.

- On page 3 line 129, the formula for how DNP was calculated is not provided. Please consider this issue.

- On page 3 line 142, “NDP” should be “DNP”. It is suggested to unify the term of “DNP”.

Author Response

This article “State-of-the-art in constructing gas-propelled dissolving mi-croneedles for significantly enhanced drug loading and delivery efficiency” introduces optimization of microparticle-embedded gas-propelled microneedles preparation technologies and formulation composition which enhances drug loading and transdermal delivery efficiency. The approach of this manuscript is considered to be an informative one; however, major revision is recommended due to the following points.

- The novelty of the manuscript is the concept of using gas-propelled microneedles which enhance local drug absorption. It is recommended to provide a figure for mechanism of pneumatic initiators which were utilized to produce carbon dioxide (CO2) bubbles through acid-base reaction in water, thus accelerating the dissolution of MNs. There is lack of data to detail the novelty of gas propelled microneedles.

Response: Thanks for your kind advice. The mechanism diagram of the pneumatic initiators is added in the revised manuscript (Figure 1) and shown as the following.

- On page 2 line 86, “Based above knowledge” should be “Based on above knowledge”.

Response: It has been corrected in the revised manuscript (Page 2, Line 89).

- On page 3 line 129, the formula for how DNP was calculated is not provided. Please consider this issue.

Response: The formula used to calculate DNP has been added in the revised manuscript (Page 4, Line 167) and was shown as the following.

- On page 3 line 142, “NDP” should be “DNP”. It is suggested to unify the term of “DNP”.

Response: The term of “DNP” has been unified in the revised manuscript (Page 4, Line 179, Page 11, Line 359/367/368/371).
